# Prostate Cancer Radiogenomics—From Imaging to Molecular Characterization

**DOI:** 10.3390/ijms22189971

**Published:** 2021-09-15

**Authors:** Matteo Ferro, Ottavio de Cobelli, Mihai Dorin Vartolomei, Giuseppe Lucarelli, Felice Crocetto, Biagio Barone, Alessandro Sciarra, Francesco Del Giudice, Matteo Muto, Martina Maggi, Giuseppe Carrieri, Gian Maria Busetto, Ugo Falagario, Daniela Terracciano, Luigi Cormio, Gennaro Musi, Octavian Sabin Tataru

**Affiliations:** 1Department of Urology, IEO, European Institute of Oncology IRCCS, 20141 Milan, Italy; ottavio.decobelli@ieo.it (O.d.C.); gennaro.musi@ieo.it (G.M.); 2Department of Oncology and Hemato-Oncology, University of Milan, 20122 Milan, Italy; 3Department of Cell and Molecular Biology, George Emil Palade University of Medicine, Pharmacy, Sciences and Technology from Târgu Mureș, 540142 Târgu Mureș, Romania; mdvartolomei@yahoo.com; 4Department of Urology, Medical University of Vienna, 1090 Vienna, Austria; 5Urology, Andrology and Kidney Transplantation Unit, Department of Emergency and Organ Transplantation, University of Bari, 70124 Bari, Italy; giuseppe.lucarelli@inwind.it; 6Department of Neurosciences, Reproductive Sciences and Odontostomatology, University of Naples “Federico II”, 80131 Naples, Italy; felice.crocetto@unina.it (F.C.); biagio.barone@unina.it (B.B.); 7Department of Maternal Infant and Urologic Sciences, Policlinico Umberto I Hospital, “Sapienza” University of Rome, 00161 Rome, Italy; alessandro.sciarra@uniroma1.it (A.S.); francesco.delgiudice@uniroma1.it (F.D.G.); martina.maggi@uniroma1.it (M.M.); 8Department of Onco-Hematological Diseases, U.O.C. Radiotherapy–Azienda Ospedaliera San Giuseppe Moscati-(AV), 83100 Avellino, Italy; mattomuto@gmail.com; 9Department of Urology and Organ Transplantation, University of Foggia, 71122 Foggia, Italy; giuseppe.carrieri@unifg.it (G.C.); gianmaria.busetto@unifg.it (G.M.B.); ugofalagario@gmail.com (U.F.); luigi.cormio@unifg.it (L.C.); 10Department of Translational Medical Sciences, University of Naples “Federico II”, 80131 Naples, Italy; daniela.terracciano@unina.it; 11Department of Urology, Bonomo Teaching Hospital, 70031 Andria, Italy; 12The Institution Organizing University Doctoral Studies (I.O.S.U.D.), George Emil Palade University of Medicine, Pharmacy, Sciences and Technology from Târgu Mureș, 540142 Târgu Mureș, Romania; sabin.tataru@gmail.com

**Keywords:** radiomics, radiogenomics, genomics, prostate cancer, molecular characterization, MRI, PET-CT

## Abstract

Radiomics and genomics represent two of the most promising fields of cancer research, designed to improve the risk stratification and disease management of patients with prostate cancer (PCa). Radiomics involves a conversion of imaging derivate quantitative features using manual or automated algorithms, enhancing existing data through mathematical analysis. This could increase the clinical value in PCa management. To extract features from imaging methods such as magnetic resonance imaging (MRI), the empiric nature of the analysis using machine learning and artificial intelligence could help make the best clinical decisions. Genomics information can be explained or decoded by radiomics. The development of methodologies can create more-efficient predictive models and can better characterize the molecular features of PCa. Additionally, the identification of new imaging biomarkers can overcome the known heterogeneity of PCa, by non-invasive radiological assessment of the whole specific organ. In the future, the validation of recent findings, in large, randomized cohorts of PCa patients, can establish the role of radiogenomics. Briefly, we aimed to review the current literature of highly quantitative and qualitative results from well-designed studies for the diagnoses, treatment, and follow-up of prostate cancer, based on radiomics, genomics and radiogenomics research.

## 1. Introduction

Prostate cancer (PCa) is the most frequently diagnosed cancer in Western Europe, the Americas, Australia, and the Central African continent, recognized as the second leading cause of death after lung cancer [1]. Only in the United States, 248,530 patients will be diagnosed with PCa in 2021, with a 5% increase compared to 2020 and 34,130 estimated deaths [2]. Based on the prostate-specific antigen (PSA) values at diagnosis, and the biopsy results and clinical stage, patients with PCa are stratified in risk categories and treated according to their disease prognosis [3]. Active surveillance (AS) is considered the gold standard treatment for patients with low-risk, indolent disease. Active treatment with radical prostatectomy (RP) or radiation therapy should be considered for patients with intermediate-risk PCa. Finally, multimodal therapies, including active treatment to the prostate and systemic therapies, may be necessary for patients harboring adverse high-risk diseases [4,5]. The main directions for all the studied cancers are those regarding the methods that guide the implementation of well-designed studies [6]. For this purpose, the image biomarker standardization initiative (IBSI) (35) will possibly lead to a consensus and a standardization of radiomics features and image processing [7]. The TRIPOD (transparent reporting of a multivariable prediction model for individual prognosis or diagnosis) statement is intended for studies to validate multivariable prediction models [8]. All studies point to a conclusion that radiogenomics is a part of the desired current precision medicine.

In the last years, the introduction of advanced imaging techniques, such as multiparametric magnetic resonance of the prostate (mpMRI) and prostate-specific membrane antigen positron emission tomography (PSMA-PET) scans, in addition to the availability of novel molecular markers, have shifted the paradigm of PCa screening, diagnosis, and treatment to a more individualized approach. According to the latest guidelines, every man at risk of PCa should undergo magnetic resonance of the prostate (MRI) prior to prostate biopsy [4,5]. If the MRI is positive, additional cores are taken from suspicious MRI lesions to improve prostate sampling. Additionally, MRI has been proven to provide higher staging accuracy compared to digital rectal examination (DRE), allowing a more conservative dissection in patients undergoing RP [9]. On the other hand, while Gleason grading and histological analysis are based on glandular architecture and the phenotypic appearance of PCa, novel techniques for the high-throughput sequencing of ribonucleic acid (RNA) and deoxyribonucleic acid (DNA) extracted from cancer cells helped to characterize PCa at a genotypic level [10]. Being the latest studied in PCa genomics, the concept of the heterogeneity of PCa, the intratumoral modifications, clonal and subclonal alterations, microheterogeneity, macroheterogeneity, the multifocal nature of PCa, and the inter-tumoral heterogeneity need to be matched between imaging and molecular pathology, for establishing the clinical implications [11,12,13].

Therefore, radiomics and genomics represent two of the most promising fields of cancer research. With advanced computational methods, it is now possible to extract quantitative features from patients scans, and to analyze the high quantity of data coming from these novel diagnostic tools to ultimately improve the risk stratification and disease management of patients with PCa. The combination of these fields, namely, radiogenomics, founds its foundation on the correlation between advanced image texture analysis, molecular characteristics, and patients’ outcomes. Radiogenomics have been studied in only a few cancers, such as glioblastoma [14,15,16,17], breast cancer [18,19,20,21], renal cancer [22,23,24], and other common neoplasms, which are summarized in a review by Shui et al. [6].

In recent years, radiomics features have been linked to the molecular characteristics of cancer tissue, genomics, proteomics, and metabolomics. This new ongoing field of research for PCa is an extension of radiomics, and its main focus is a tailored approach in the diagnosis of aggressive PCa [25], and the prediction of prognosis [26], progression [27], and treatment response [25]. The term radiogenomics may be correctly referred to in two different types of studies. Those who evaluate the correlation of the imaging quantitative features and molecular characteristics of visible and non-visible cancer foci, and those who aimed to develop radiogenomic models to predict disease outcomes, combining complementary information coming from radiomics and genomics [28,29].

The aim of this review was to summarize the current evidence regarding radiogenomics and its application in patients with PCa, and to give an overview of the current evidence and future directions of radiogenomics. This will emphasize the role of the present application of radiogenomics in clinical settings, with the need to better understand the combination between radiomics and genomics in PCa development, detection, treatment, and follow-up of PCa patients, to better adapt the management of these patients.

## 2. Results

A total of 1066 research papers were identified. Fourteen duplicate files were removed and 1052 abstracts were screened for eligibility. One hundred and twenty-five met the criteria for quantitative analysis using radiomics, radiogenomics, PCa, genomics, MRI, PET-CT as the keywords (Figure 1). After full-text examination, a total of 61 papers were ultimately included.

### 2.1. Radiomics

Radiomics is the extraction of the quantitative image analysis of textures and features (region size, shape or location, histogram of volume intensity, texture analysis, transform analysis, fractal analysis) [30], provided by imaging tools (e.g., mpMRI) that focus on the improvement of the analysis of large datasets through semi-automatic or automatic software [31,32], with the aim of pinpointing the localization of the PCa and assessing its aggressiveness [33,34]. These models were already studied in a variety of cancers [35,36,37,38,39]. The recent rise in artificial intelligence (AI) and machine learning (ML) algorithms has introduced new classifications for PCa, regarding the differentiation of favorable from unfavorable disease [40,41]; the quantitative assessment of information predicting the tumor Gleason score [31,32,42,43,44] and biochemical recurrence (BCR)-free survival [45]; the identification of tumors through mpMRI [43,46]; the development of new detection features, such as advanced zoomed diffusion-weighted imaging (DWI) and conventional full-field-of-view DWI [47]; texture analysis of prostate MRI in the prostate imaging reporting and data system (PIRADS) for PI-RADS 3 score lesions [48]; the creation of frameworks for automated PCa localization and detection [49]; and, finally, the management of radiotherapy treatment and toxicity [50,51,52,53,54,55,56], and the prediction of BCR [57,58,59,60,61,62,63,64,65]. Additionally, radiomics and AI algorithms will help to limit the discrepancies between different readers [66]. Indeed, mpMRI of the prostate, which has gained popularity as the most reliable imaging technique for PCa diagnosis and treatment, provides qualitative and quantitative parameters. The qualitative aspect is linked to the ability of an expert radiologist to provide an accurate scoring for the lesion images in the prostate tissue [26]. Quantitative measurements, such as tumor size, prostate volumes, and radiomics features are computed directly from the image [67] and could be considered reader-independent. A summary of specific medical expressions is accessible in Table 1.

Despite the reported advantages, challenges remain in deeply identifying the prognostic and predictive factors for individual patients, developing markers to tailor the diagnosis and treatment of low-risk and high-risk PCa patients.

### 2.2. Radiomics in Prostate Cancer Management

Although it is beyond the purpose of this review to focus on the radiomics technical terms, we have briefly provided a reminder of them because they represents the starting point in this research field. Articles that studied radiomics in PCa are just briefly reminded and not analyzed in detail, and we had them incorporated in a table, along with their clinical outcomes and results. In PCa, the use of radiomics aids prostate volume selection and segmentation [30,40,46,73,74,75,76], PCa screening [28,77,78], detection and classification [29,77,79,80,81], in addition to its role in risk stratification [61,76,82,83], treatment [59,75,78,84,85,86], and prognosis. One of the first studies that analyzed the imaging features for PCA was performed by Khalvati et al. [87], with the goal of creating a radiomics-based auto detection method utilizing an mpMRI feature model that combined computed high b-value DWI (diffusion-weighted imaging) and correlated diffusion imaging, which was then evaluated through a support vector machine (SVM) classifier. The study reported good results in terms of sensitivity (95% CI 0.76–0.91), specificity (0.86 95% CI 0.82–0.91), and area under the curve (AUC) (0.90 95% CI 0.88–0.93). mpMRI-based radiomic features need, however, to still be largely tested, in order to assess the robustness and reproducibility of methods and workflow; therefore, a proper standardization of MRI image acquisition across institutions should be encouraged as an initial step. [75]. Similarly, more data are required from clinical trials to accurately distinguish cancerous versus benign lesions, to assess the robustness of radiomics-based predictive models, and to standardize features as the automatic segmentation of gross tumor volume [50,88,89,90]. In the detection of clinically significant PCa, the combination of radiological and clinical radiomic models was, indeed, among the best methods to predict clinically significant PCa in patients with a PIRADS score of three or more. The development of different models in an automatic fashion, using ML and AI techniques, and the construction of nomograms [91] could further improve the radiomic potential on this issue. The currently existing data are promising, with radiomics outperforming PIRADS v2 in the detection of high-grade versus low-grade PCa, although some limitations remain regarding the standardization of data, and further studies are required to confirm the performance of radiomics compared to conventional radiological analysis [92]. Moreover, radiomics models are useful in the detection of prostate extracapsular extension (ECE), and allows predictive models to be build for the pretreatment detection of ECE, focusing on a combined model of clinical, conventional radiology and radiomics [93,94,95].

In Table 2, we incorporated the details of the current research on the potential of radiomics to detect PCa, differentiating between aggressive and indolent disease, and ECE, reporting clinical outcomes of interest, accuracy, and imaging modality.

### 2.3. Genomics and Molecular Tumor Characterization

Genomics and molecular characterization permit the detection and characterization of PCa, improving diagnostic and prognostic accuracy. The requirement of tissue samples through biopsy, however, limits the clinical application in everyday care [112]. Another limitation of tissue sampling is related to the presence of tumor heterogeneity. From a clinical, morphological, and molecular point of view, PCa is indeed a highly heterogeneous disease. The tissue obtained via prostate biopsy could therefore lead to a biased assessment of the samples, missing out relevant scorings and gradings of cancer [27]. The known multifocality of prostate cancer suggests the involvement of multiple genes with different clonal origins. Multiple foci in the prostate gland could, therefore, harbor different cancers. Genetic profiling of PCa aims to correlate those changes in different genes expression, with oncological outcomes, in order to achieve an improved understanding of different clonal origins, and to improve diagnostic and therapeutic processes [11,113].

Genomic biomarkers, validated as independent predictors of oncological outcomes, are currently being used more and more in clinical practice, in the process of decision making of PCa patients [114]. The following four available genomic biomarkers are approved and available: Oncotype Dx test^®^ [115], Prolaris test^®^ [116,117,118,119], and Decipher test^®^ [120,121,122,123]. Another investigated tissue biomarker, the mutated tumor suppression gene phosphatase and tensin homolog (PTEN) was assessed in prostate cancer radiogenomics [124], along with whole-exome DNA (deoxyribonucleic acid) sequencing data [125].

#### 2.3.1. Genomic Risk and Molecular Imaging in Prostate Cancer

mpMRI has been validated as a radiologic technique used for PCa detection, targeted biopsy, and for better surveillance and staging of the disease [126]. AS, for low-risk PCa, is offered with the intent of reducing treatment-related events, but 30% of patients [4,127] are upstaged with the help of mpMRI-targeted biopsies [128]. Still, 10 to 20% of clinically significant PCa’s are not visible by mpMRI [129]. The pathological, molecular, and micro-environmental hallmarks are poorly understood in PCa [130]. Aggressive PCa seems to have genomic alterations [131], and these molecular expressions, along with mpMRI phenotypes, are likely to have a prognostic significance [132]. Parry et al. [133] used low-pass whole-genome, exome, methylation, and transcriptome profiling of patient tissue cores from the same glands, along with circulating free and germline DNA from patients serum, in order to analyze the genomic, epigenomic, and transcriptomic images that are visible, or not, on mpMRI in PCa clinically localized disease. From the analyzed cores, 27% (six tumors) were not visible on mpMRI, and three (50%) cores that were harvested from non-visible tumors on mpMRI had one or more genetic alterations spotted in metastatic castration-resistant PCa [134,135]. Radtke et al. [136] aimed to fuse the mpMRI imaging with a multi-dimensional map of biopsies and genomic features, to compare the genomic signals from the biopsy site, and from surrounding and other benign spots in the same prostatic gland in patients with RP. A strong association was observed between PI-RADSv2 and the Decipher test^®^, and the genomic Gleason grade classifier score, and the combination between targeted fusion biopsy and genomics showed a very good correlation with RP and genomics [137]. These studies are summarized in Table 3 and discussed below.

##### Prostate Cancer Antigen 3 (PCA3)

PCA 3 is an mRNA expression analysis of patients who are suspicious of having PCa, with a negative prior biopsy, from the post-DRE urine sample [151]. Researchers aimed to evaluate the combination between MRI and PCA3 in different settings. De Luca et al. [138] determined, in 282 patients with a negative prior biopsy, the association of PCA3 score, PI-RADS grade, and Gleason score, undergoing MRI/TRUS fusion-targeted biopsy, finding a statistically significant association between PCA3 score and PIRADS grade groups 3, 4, and 5 (*p* = 0.006). Alkasab et al. [139] looked at the potential of combined PCA3 and mpMRI, in PCa patients with two negative biopsies. The results were limited, with a positive PCA3 associated with high-grade PCa at the final pathology (*p* = 0.0435), but not with an overall PCa diagnosis (*p* = 0.128) and a positive PCA3 associated with high-grade PCa at the final pathology (*p* = 0.0435). The combination of PCA3 and mpMRI in 187 patients with no prior prostate biopsy, published by Fernstermaker et al. [140], found that PCA3 is associated with an MRI suspicion score of two and three (*p* = 0.004), but not four and five (*p* = 0.340), with roughly no significant addition in cancer diagnoses. Perlis et al. [141] analyzed a cohort of 470 men with mpMRI and PCA3, and identified that the PIRADS score and PCA3 score were independently associated with clinically significant PCa on a second biopsy. Clinically significant PCa on the biopsy was not identified in patients with negative mpMRI and a normal PCA3 score, with a negative predictive value of 100% (*p* < 0.0001).

##### Decipher Test^®^

Decipher^®^ is a clinical–genomic risk grouping system, consisting of the analysis of 22 RNA markers that were originally obtained from radical prostatectomy samples and, lately, from prostate biopsy, to predict mortality and metastasis [123]. Martin et al. [142] explored the association between PIRADS v2 score, histological grade, and Decipher^®^ score in biopsy samples from low- and intermediate-risk PCa patients, finding an association between the Decipher^®^ biopsy genomic test and Gleason grade group, independently from the PIRADS v2 score. In a larger trial, by Falagario et al. [143], Decipher^®^ test and mpMRI were analyzed, in order to better define a favorable intermediate-risk PCa in a cohort of 509 patients, reporting multivariable analysis, unfavorable intermediate risk category (*p* < 0.001), and Decipher test^®^ (*p* = 0.012) as statistically significant predictors of adverse pathology, while mpMRI did not achieve statistical significance (*p* = 0.059). Similarly, Jambor et al. [144] explored the use of a routine clinical prostate mpMRI and Decipher^®^ genomic classifier score to predict biochemical recurrence in 91 patients who underwent radical prostatectomy (of which 48 developed biochemical recurrence), concluding the absence of improvement in the predictive performance of both tests combined, compared to individual utilization. Beksac et al. [145] retrospectively analyzed the association of Decipher^®^ score, which was significantly correlated with lesion size (*p* = 0.03), PIRADS score (*p* = 0.02), and extraprostatic extension (*p* = 0.01), reporting, in addition, increased activity of the PI3K-AKT-mTOR, WNT-b, and E2F signaling pathways in PIRADS 5 lesions, and of estrogen and inflammation/stress (NFkB and UV response) pathways in PIRADS 4 lesions. Moreover, in research by Purysko et al., it was found that MRI-visible lesions had higher Decipher^®^ scores than MRI-invisible lesions (*p* < 0.0001), but some lesions were still classified as intermediate/high risk by Decipher^®^ and were not identifiable by mpMRI. This suggests that Decipher^®^ added on to MRI will probably not lead to significantly more detections of cancer. Conversely, despite technical advancements in prostate imaging, such as MRI, not appearing to be superior to Decipher^®^ testing in the prediction of adverse pathology, recently, Li et al. [45] proposed a new imaging-based nomogram to predict biochemical recurrence and adverse pathology, reporting promising results with an AUC (0.71, 95% CI 0.62–0.81) higher than Decipher^®^ AUC (0.66, 95% CI 0.56–0.77) and prostate cancer risk assessment (CAPRA) score AUC (0.69, 95% CI 0.59–0.79).

##### Oncotype Dx Test^®^

The Oncotype Dx test^®^ prostate cancer assay includes 5 reference genes and 12 cancer genes, and it was validated using predefined acceptance criteria [152], to predict PCa aggressiveness [115]. Leapman et al. [147] aimed to evaluate the association between mpMRI findings and a biopsy-based RT-PCR assay comprised of a 17- gene (Oncotype Dx test^®^) genomic prostate score, among men with clinically favorable PCa, following the initial diagnosis. The results show that genomic prostate score differences were reported among MRI categories for patients with the Gleason pattern 3 + 4 (*p* = 0.010), but not for the Gleason pattern 3 + 3, while no differences were reported among androgen signaling or proliferation genes. Salmasi et al. [148] investigated the ability of the genomic prostate score to predict adverse pathology in 134 patients undergoing MRI-guided prostate biopsy, resulting in the multivariate analysis that confirmed that the genomic prostate score is a significant predictor for adverse pathology (*p* < 0.001).

##### ConfirmMDx^®^

ConfirmMDx^®^ is a tissue-based gene assay that screens for epigenetic modifications identified in a prostate tissue sample. Alterations in DNA methylation in tumor suppressor genes (GSTP1, GASSF1, and APC) are identified by this assay, with the aim to stratify the risk of men with prior negative biopsies [153,154]. An original research by Artenstein et al. [149], using mpMRI PIRADS score lesions after ConfirmMDx^®^ sampling, indicates that a negative ConfirmMDx^®^ test was somehow in accordance with negative MRI results (71.4%). In addition, PIRADS 5 lesions were identified in the anterior base of the prostate, confirming the usefulness of ConfirmMDx^®^ sampling in a fusion-targeted biopsy setting.

##### Prolaris Test^®^

Prolaris test^®^ is a 46-mRNA genomic test that analyzes prostate biopsy tissue [155]. It generates a risk score that has been associated with BCR, metastasis, and cancer-specific survival in PCa patients [156]. To date, one significant study has been published, from Wibmer et al. [150], which has studied the associations between MRI and the expression levels of cell cycle genes. In the prostatectomy subgroup, ECE on MRI (*p* ≤ 0.001–0.001) and cycle genes risk scores (*p* = 0.049) were significantly associated with the Gleason score 4 + 3 or higher, ECE, and lymph node metastases.

The available data on combining genomics and imaging show that PCA3 and MRI features are limited, and with little evidence on the actual comparative research between genomic risk tests and MRI. The results point out the fact that there is a real potential in combining PCA3 and MRI scores. The Decipher genomic score is good at predicting the Gleason grade group and adverse pathology [157], but the combination with mpMRI could not improve the predictive performance of BCR; a wide and overlapping distribution of GPS results were observed across PIRADS scores in some studies, and only one study showed an association with PIRADS score. For ConfirmMDx^®^ test and ConfirmMDx^®^, only two studies showed that in the RP subgroup, some MRI features and cycle genes risk scores were associated with clinically significant PCa, and ECE and lymph node metastases, and that a negative ConfirmMDx^®^ test is in accordance with negative MRI results, respectively. Most of the studies are of retrospective design, but to determine the potential ability of combining genomics with imaging, in improving PCa diagnosis, there is a need for well-designed randomized controlled trials.

#### 2.3.2. Radiogenomics in Prostate Cancer Management

In PCa research, several papers focused on the differential expression of genomic markers in MRI-visible and -invisible lesions. One of the first experiences with radiogenomics was reported by McCann et al., who performed a retrospective analysis of 30 patients with proven PCa at biopsy and MRI performed prior to RP [124]. The aim was to investigate associations between the quantitative imaging features of multiparametric MRI and the PTEN expression of PCa. They found a correlation between Gleason score and PTEN expression (r = −0.30, *p* = 0.04), and between k_ep_ and PTEN expression (r = −0.35, *p* = 0.02).

Stoyanova et al. [158] reported quantitative mpMRI features and gene expression in biopsy tissue. The authors introduced the concept of habitat, which is a combination of images from multiple modalities, compiling pieces of orthogonal information. In radiogenomic analysis, genes were significantly associated with radiomic features (*p* < 0.05). This was the first study that correlated radiogenomic parameters with prostate cancer in men with MRI-guided biopsy.

In another work by Renard-Penna et al. [119], prognostic biomarkers were identified through radiogenomics, with a Gleason score > 3 associated with a longer median tumor diameter and a lower ADC (both *p* < 0.0001). The authors also found an association between Prolaris^®^ cell cycle progression score, Gleason score (r = 0.199, *p* = 0.04), and PIRADS score (r = 0.26, *p* = 0.007). This paper states that mpMRI is able to predict low- and high-risk Gleason scores in the tumor, and suggests that the management of the early stages prostate cancer could strongly benefit, by performing MRI-targeted biopsy coupled with molecular analysis.

Jamshidi et al. [125] performed a research study where a multi-region spatial map was created with mpMRI images and histopathology of the prostate gland, after RP combined with whole-exosome DNA sequencing, performed on the regions of interest. No statistically significant linear correlation was identified between individual mutations and mpMRI imaging parameters or PIRADS scores (*p* = 0.3). This article is one of the few that have performed MRI and whole-exome sequencing. It shows a continuum of mutations across regions that were found, via histologic analysis, to be high grade and normal.

Houlahan et al. [130] identified small nucleolar RNAs that were significantly more likely to have elevated abundance in visible tumors (odds ratio (OR) 4.4; FDR = 0.002; Fisher’s exact test). Two small nucleolar RNAs (snoRNAs) that were identified (SNORA37 and SNORA12) were prognostic; a high abundance was associated with early BCR in an independent intermediate-risk PCa cohort (hazard ratio (HR) 2.00 and 2.00; *p* = 0.053 and 0.051). Another interesting finding was that a snoRNA signature accurately predicted PIRADS v2 score of 5 for PCa tumors, with 76% accuracy. Noncoding transcripts were associated with mpMRI visibility. The authors introduced a new term, nimbosus, characterized by the combination of pathological, molecular, and micro-environmental events, including intraductal carcinoma and cribriform architecture, genomic instability, SCHLAP1 expression, and hypoxia. The signature of snoRNAs associated with nimbosus hallmarks seems to have the potential to differentiate visible from non-visible tumors. This paper observed that MRI findings are associated with the biological features of aggressive prostate cancer.

Li P et al. [159] also investigated the visibility of tumors on MRI and their biology, and identified four genes (PHYHD1, CENPF, ALDH2, GDF15) that predict MRI visibility (AUC: 0.86) and progression-free survival (in the following two external datasets: GSE21034 and GSE40272 genes). The four genes define two groups with significantly different BCR-free survival (HR = 2.53 (1.55–4.11), *p* < 0.001, and HR = 1.3 (1.04–1.63) *p* = 0.021, respectively), concluding that MRI visibility was associated with the genetic features that were linked to poor prognosis. This article looked at the genes involved in PCa prognosis and metastasis, indicating that MRI visibility has prognostic significance and is linked to poor prognosis.

Eineluoto et al. [160] determined the association between PTEN and ETS-related gene (ERG), with visible and invisible PCa lesions on MRI. A retrospective analysis of 346 patients with pre-RP MRI, PTEN and ERG tissue microarray staining, was performed. Patients with MRI-invisible lesions had less PTEN loss and ERG-positive expression compared with patients with MRI-visible lesions (17.2% vs. 43.3%, *p* = 0.006; 8.6% vs. 20.0%, *p* = 0.125). This study shows that PTEN loss, BCR, and non-organ-confined disease were more often encountered with MRI-visible lesions.

Hectors et al. [161] retrospectively analyzed a cohort of 64 patients to evaluate the predictors of the final pathology Gleason score. Several MRI radiomics features, based on both T2w and DWI sequences, were found to be significantly associated with the pathological Gleason score, prognostic gene expression signatures, including Decipher^®^, and 698 PCa-related gene expression levels. Machine learning was used to develop a model to predict a Gleason score of 8 or greater, with a fair performance (AUC 0.72), and excellent performance to predict a Decipher^®^ score of 0.6 or greater (AUC 0.84). This study found 14 MRI imaging radiomics features correlated with Gleason score.

Li L et al. [162] evaluated radiomic feature-derived MRI T2w and ADC maps of the prostate, to distinguish different Decipher^®^ risk groups (low, intermediate, and high). Their model outperformed the prediction using PIRADS v2 (AUC = 0.67), but showed comparable performance with the Gleason grade group (AUC = 0.80), and the best discriminating radiomic features were correlated with gland morphology and gland packing on corresponding histopathology (R = 0.43, *p* < 0.05). Sun et al. [163] studied full transcriptome genetic profiles that were obtained using next-generation sequencing and texture features (obtained from T_2_w images and parametric maps from functional mpMRI). Immunohistochemistry identified only a weak association between mpMRI features and hypoxia gene expression (*p* < 0.05). This study proposed a model comprised of radiomic features derived from T2 and ADC images, to distinguish different Decipher risk groups, and it outperformed the risk prediction of PIRADS v2.

Fischer et al. [27] studied a radiogenomic model including clinical, imaging, and genomic (gene and miRNA expression) datasets for 298 PCa patients. Four biomarkers (Alanyl membrane aminopeptidase, microRNA-mir-217, mir-592, mir-6715b) were found to be able to differentiate between the T2c and T3b PCa stages, which were highly correlated (average r = ±0.75) with aggressiveness on related radiomics imaging features. This research proposed a model that found that a radiogenomic approach using four biomarkers can improve the prediction accuracy for disease stage and the characterization of PCa aggressiveness.

Wibmer et al. [150] analyzed, retrospectively, the association of cell cycle risk score (Prolaris^®^ test) and PIRADS v2 score, ECE, and quantitative metrics. Patients with ECE on their MRI had a significantly higher mean cell cycle risk score (reader 1: 3.9 vs. 3.2, *p* = 0.015; reader 2: 3.6 vs. 3.2, *p* = 0.045). This paper found that the radiomic phenotypes of ECE on MRI indicate a more aggressive genotype of PCa.

VanderWeele et al. [164] investigated the risk of aggressive PCa prior to prostatectomy, using a radiomic model to assess the immunohistochemical analysis of cells expressing PTEN, obtaining two perfusion imaging contrast uptake parameters that mathematically correlated with PTEN expression (r = 0.25, *p* < 0.1 and r = 0.43, *p* < 0.01), and T2w unevenness also showed some correlation tendency (r = −0.25, *p* < 0.1). This preliminary article suggests that a fast contrast uptake of cancer on DCE-MR imaging and a T_2_w imaging feature are potentially associated with prostate cancer PTEN expression.

Switlyk et al. [165] investigated PTEN expression in PCa patients. Forty-three patients who underwent pre RP MRI were included. Based on bead arrays (*p* = 0.006) and real-time quantitative polymerase chain reaction (RT-qPCR) (*p* = 0.03) data, a significantly lower ADC, derived from DWI, was found in tumors with low PTEN expression. ADC was negatively correlated with Gleason score (*p* = 0.001) and tumor size (*p* = 0.023). This article found that in aggressive PCa, due to PTEN loss, ADC derived from DWI may be useful in detecting these patients.

The summary articles with the molecules studied and imaging performed, and the methodology and results, are available in Table 4.

## 3. Discussion

Radiogenomics has been thoroughly studied in prostate cancer, with investigations between quantitative image features and single gene expression, which delivered promising results. In particular, regarding the characterization of PTEN expression, a weak, but significant, association has been reported between imaging features and the Gleason score of a peripheral zone PCa [124]. Similarly, a significantly lower ADC (negatively correlated with Gleason score and tumor size) was found for tumors with low PTEN expression, which was, in addition, negatively correlated with lymph node involvement [165]. Another study showed that imaging uptake parameters reported a mathematical correlation with PTEN expression (r = 0.25, *p* < 0.1 and r = 0.43, *p* < 0.01, and T2w unevenness also showed some correlation tendency (r = −0.25, *p* < 0.1) [164]. Other studies also correlated radiomic features with Gleason score and PIRADS sum score [119]. The development of genome sequencing studies looked at the genomic profile, with the help of radiomics, in order to investigate broader aspects of the genomics potential, while earlier research studied a small number of patients and a small number of genes [158]. Radiogenomic models can determine the gene expression profiles from biopsy samples. In early studies, one gene was selected to be studied [166]. Lately, genomic research showed that gene expression is not influenced much by sampling tumor heterogeneity [167]. Retrospective articles that classified gene expression in low- or high-risk scores, using the Decipher^®^ genetic risk profile, could predict a Gleason score of 8 or greater (AUC 0.72) and a Decipher^®^ score of 0.6 or greater (AUC 0.84), and had comparable performance with the Gleason grade group (AUC = 0.80), but these are modest results. Some of the results of radiomic studies can distinguish the genomic signatures associated with high-risk disease [130] and hypoxia gene expression [34]. A study reported that quantitative mpMRI features and gene expression in biopsy tissue and in radiogenomic analysis genes were significantly associated with radiomic features [158], while the other identified biomarkers were able to differentiate between the T2c and T3b prostate cancer stages [27]. Some retrospective radiogenomic research identified those visible mpMRI lesions, and the genes that can predict visibility, and allows the identification of high risk patients of harboring aggressive disease [130,159,160]. Research being performed on the association between prostate MRI and tissue-based gene expression shows that genomic testing can reveal more about the disease biological processes [168,169,170]. These two possibilities can enhance the pace of monitoring the patients, especially in active surveillance, which can indicate those at greater risk of harboring aggressive disease, and could potentially be more strictly followed after diagnoses. Finally, the combination of MRI and the genomic test could raise the health-care systems burden [171]. At this point, radiogenomics is an emerging field that studies the correlation between image phenotypes and genomics inside a tumor. The translation from clinical studies into clinical practice is still a challenge, because matching data from imaging and whole-genome sequencing is very complex. The different imaging techniques and machines used to provide imaging data make it difficult to allow standardized results. The low number of patients included in the radiogenomic research also limits the validity of the results. Some limitations of this review have to be addressed. Only a few databases and online search engines were used to retrieve relevant data. The search for the above topic was performed manually, by two independent researchers. Scrutinizing more databases and registries, using a software search, may have provided more relevant data for the subject.

### Current Challenges/Limitations and Future Perspectives

Radiogenomics holds great promise, but it is a new area of research, which means that it is, therefore, facing several limitations [172]. In the last 5 years, only a few studies have been published on radiogenomics in PCa, and, up to date, no application in the clinical setting has been properly established. Even in this case, this is due to several limitations associated with radiogenomic analysis [173].

Radiomic flow is a complex process, and every aspect of the image acquisition, such as defining and contouring the regions of interests, and choosing the best features to be extracted and the proper statistics to be applied, remains challenging. Lately, explainable AI (XAI), using DNNs (deep neural networks), may help radiomics in classification and prediction in the clinical setting [174,175], and a controllable and explainable probabilistic radiomics framework was proposed, through which a 3D CNN feature is extracted upon the lesion region only and is used to approximate the ambiguity distribution over human experts [176]. These new features will have to be further validated. The heterogeneity that limits the standardization of the results obtained, and applying it for every scenario, leads to a domain shift, which represents the difference between the training data distribution and the distribution of where a model is employed [175]. Currently, there are methods that try to solve the problem of domain shift by using domain adaptation and fine-tuning methods [177,178]. A significant time is required for experienced radiologists to obtain a proper manual delineation, and the inter-observer variability in reading and segmenting regions of interest represent the major drawback. Similarly, the different protocols designed to improve the specificity and sensitivity of radiomic features performed in the studies evaluated, without appropriate standardization, could further limit the results obtained [179,180]. The utilization of different acquisition protocols, scanners, and radiomic studies represent another risk, which could compromise the results and predictive performances, due to the presence of random errors/noise occurring in excessively complex models (such as having too many parameters or forcing a linear model for non-linear data). To further complicate the research, gene expression and signaling pathways [181] are intrinsically and extremely complex. Matching the data from whole-genome sequencing with imaging data is difficult, due to the large amount of data and features, and due to the differences in patient characteristics and imaging protocols. The small patient cohort and retrospective nature of radiogenomic studies represent another limit to the standardization of the protocols [179]. However, despite those difficulties, radiogenomics is trying to overcome the known heterogeneity of PCa with a non-invasively radiological assessment of the whole specific organ [26].

The future of radiogenomics could be its integration into everyday clinical practice, if larger prospective, multicenter studies and protocol standardization of the imaging features extracted will be performed, permitting the validation of the potential of this technique in the identification of relevant imaging biomarkers. The possibility to access large public databases of imaging and genome data will further ease this process [182,183]. The International Cancer Genome Consortium (ICGC), for example, provides the information of genomic, transcriptomic and epigenomic abnormalities, and somatic mutations in prostate cancer [184,185,186,187]. Some patients from these genome databases have both radiomic and genomic data [83,188]. Lately, the AI and big public databases with genomics and imaging features could be used to develop CAT, to ease the translation of results into clinical practice and to aid in the clinical decision. This also comes with a number of drawbacks, as follows: a lack of standardization, and imaging and reporting protocols that differ significantly among institutions. The process of contouring the regions of interest is mainly performed in a manual/classic fashion, and the user inter-variability is unfortunately still high. Automatic and semi-automatic computer designed software has been proposed to overcome these pitfalls [189]. All the present studies utilized conventional radiomic features [27,165] and, in the future, adding deep learning techniques may improve the results. By implementing models to label the region of interest, using deep learning methods, with the help of clinicians to establish the ground truth, would probably improve the performance of large-scale datasets from genomic and imaging databases [190,191,192,193]. A summary of the advantages and limitations of radiogenomics is listed in Table 5.

## 4. Material and Methods

A literature search was performed in June 2021 using PubMed, Google Scholar, and Web of Science. A free text strategy was deemed to be most suitable for the following purpose: “Radiomics and PCa and Genomics and Radiogenomics and MRI”. We focused on papers published in the last five years. After literature search and duplicates removal, the abstract of each study was assessed to evaluate the eligibility. Finally full texts of selected articles were retrieved and screened. Additional papers were included by reference lists if deemed appropriate. In this literature review we had included the radiomics data grouped in a summary table in order to emphasize the early importance of this research, radiogenomics data (MRI and PET-CT), including data from leading studies later than five years, genomic tests, commercially available and novel genetic and transcriptomic [197] and metabolomic [198] biomarkers studied in conjunction with imaging.

## 5. Conclusions

We had identified a lot of blank space in the radiogenomics literature, when it comes to research in prostate cancer. No prospective randomized control trials were published. At this moment, we do not have any utility or validity for the use of radiogenomics in clinical practice. Many gaps remain to be filled, probably some of them by using models that consist of clinical, radiomic, and genomic biomarkers, combined or alone, to improve predictive capacities. The rise in AI in medicine, especially deep learning techniques, could address those limitations and permit the clinical implementation of radiogenomics.

## Figures and Tables

**Figure 1 ijms-22-09971-f001:**
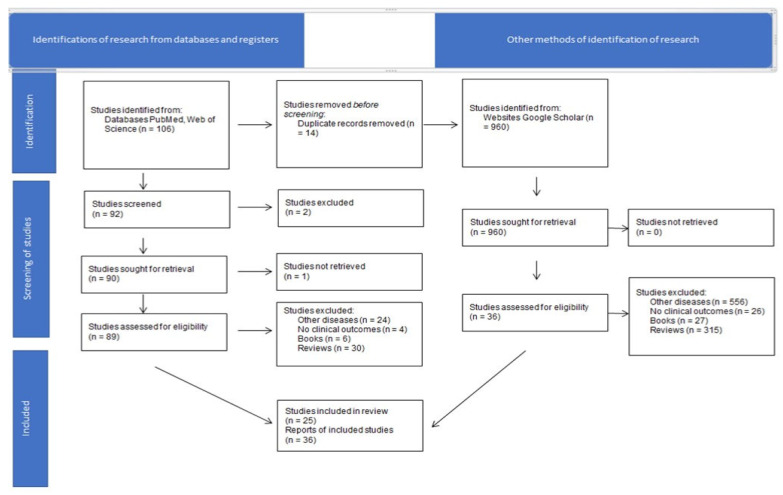
Research flowchart for identification of related articles.

**Table 1 ijms-22-09971-t001:** Short definitions for specific medical terms.

Terminology	Short Definition
Radiomics	Quantitative approach to medical imaging, enhancing existing data through mathematical analysis [68].
Genomics	Study of whole genomes, including elements from genetics. Genomics uses a combination of recombinant DNA, DNA sequencing methods, and bioinformatics to sequence, assemble, and analyze the structure and function of genomes [69,70,71].
Radiogenomics	Genomics information that can be explained or decoded by radiomics and to develop methodology to create more-efficient predictive models [72].

DNA = deoxyribonucleic acid.

**Table 2 ijms-22-09971-t002:** Clinical results of radiomics studies.

Author	Clinical Outcomes	Type of Image Acquisition	Results
Radiomics in diagnosis and detection of prostate cancer			
Zhang et al. [29]	Upgrading Gleason score from biopsy to RP	MRI	AUC: combined clinical and radiomics model 0.910, clinical model 0.646, radiomics model 0.868
Dulhanty et al. [80]	Detection of PCa	MRI	Zone-discovery radiomics model (AUC 0.86) > clinical heuristics model (AUC 0.79)
Bagher- Ebadian et al. [79]	Detection of dominant intraprostatic lesions and normal tissue	MRI	Comparison between conventional model and artificial neural network model, AUC model (0.94 and 0.95, respectively)
Qi et al. [77]	Detection of PCa through radiomics in prostate cancer patients with PSA range 4–10 ng/mL	MRI	Combined model (radiomics signature and clinical radiological risk factors) AUC 0.933, *p* < 0.05
Chen et al. [81]	Diagnosis of intermediate-/high-grade (GS ≥ 7) tumors	MRI	Radiomics-based model > PIRADS v2 model in PCa detection vs. no PCa (AUC 0.999). Validation in differentiating high-grade from low-grade PCa (AUC 0.777)
Khalvati et al. [87]	Detection of PCa	MRI	Specificity used as performance evaluation criteria can maximize the results for AUC (0.90), which leads to balanced results for sensitivity and specificity; 0.84 and 0.86, respectively
Hu et al. [47]	Detection of PCa	MRI	Mixed model better compared with mp-MRI signatures and clinically independent risk factors alone (AUC 0.81, 0.93, and 0.94 in training sets, and 0.74, 0.92, and 0.93 in validation sets, respectively)
Brunese et al. [96]	Gleason score prediction	MRI	Gleason score prediction equal to 0.98473, 0.96667, 0.98780 and 0.97561 for, respectively, Gleason score 3 + 3, Gleason score 3 + 4, Gleason score 4 + 3 and Gleason score 4 + 4 prediction
Radiomics and detection of clinically significant prostate cancer			
Wang et al. [97]	Detection of clinically significant PCa Gleason score ≥ 7 (3 + 4). Lesions defined as volume > 0.5 cm^3^ on histopathology.	mp-MRI	PCa vs. normal PZ + TZ Combined: 0.978 (0.947–0.993) PCa vs. normal PZ Combined: 0.983 (0.960–0.995) PCa vs. normal TZ Combined: 0.968 (0.940–0.985)
Kwon et al. [98]	Detection of clinically significant PCa Gleason score ≥ 7 (3 + 4)	MRI	AUC 0.82 (random forests), 0.76 (CART), and 0.76 (adaptive LASSO)
Parra et al. [99]	Detection of clinically significant PCa Gleason score ≥ 7 (3 + 4)	mpMRI	The trained models had an AUC of 0.82 and an AUC of 0.82 on validation cohort
Penzias et al. [100]	Detection of high-grade PCa	MRI	Gabor texture features identified as most predictive of Gleason grade on MRI (AUC of 0.69)
Cuocolo et al. [40]	Detection of clinically significant PCa Gleason score ≥ 7 (3 + 4)	MRI	Multivariable analysis of T2W and ADC-derived SAVR: AUC 0.78
Giambelluca et al. [48]	Presence of clinically significant PCa Gleason score ≥ 7 (3 + 4) in PIRADS 3 images	MRI	AUC of 0.769 and 0.817 on T2w or 0.749 and 0.744 on ADC maps images Analysis was performed using the GLM regression. To strengthen the reliability of the results and avoid over-fitting, 10-fold cross-validation was performed
Min X et al. [101]	Detection of clinically significant PCa Gleason score ≥ 7 (3 + 4)	mpMRI	Logistic regression modeling yielded AUC 0.872 in the training cohort and 0.823 in the test cohort
Brancato et al. [102]	Gleason Score ≥ 6 in PIRADS 3 images and in peripheral PIRADS 3 upgraded to PIRADS 4 images	MRI	PIRADS 3 images: sensitivity, specificity and accuracy (0.8, 0.51, 0.71, respectively) with AUC = 0.76. For upgraded PIRADS 4: AUC—0.89, sensitivity—0.87, specificity—0.62 and accuracy—0.82
Hou et al. [103]	Detection of clinically significant PCa Gleason score ≥ 7 (3 + 4)in PIRADS 3 lesions	mpMRI	AUC model one is 0.89 and higher than that of model two with AUC of 0.87 (*p* = 0.003)
Zhang et al. [104]	Differentiation between clinically significant PCa Gleason score ≥ 7 (3 + 4) from insignificant prostate cancer	MRI	Combination AUC of 0.95 (training group), 0.93 (internal validation group), and 0.84 (external validation group). *p* < 0.001
Gong et al. [105]	Detection of clinically significant PCa Gleason score ≥ 7 (3 + 4)	MRI	Combined clinical and radiomics model (T_2_w/DWI) yielded an AUC of 0.788
Woźnicki et al. [76]	Prediction of clinically significant PCa Gleason score ≥ 7 (3 + 4)	mpMRI	The model combining radiomics, PIRADS, PSA density and DRE showed a significantly better performance compared to ADC for clinically significant prostate cancer prediction (AUC = 0.571, *p* = 0.022)
Bernatz et al. [106]	Discriminating clinically significant PCa Gleason score ≥ 7 (3 + 4) versus indolent disease	mpMRI	Three classification models were trained and a subset of shape features improved the diagnostic accuracy of the clinical assessment categories (maximum increase in diagnostic accuracy ΔAUC = +0.05, *p* < 0.001)
Gugliandolo et al. [43]	Predictive of Gleason score, PIRADS v2 score, and risk class	mpMRi	Gleason score, PIRADS v2 score, and risk class; AUC 0.74 to 0.94
Krauss et al. [73]	PSA level in patients with low suspicion for clinically significant PCa Gleason score ≥ 7 (3 + 4).	MRI	Five radiomic features were significantly correlated with PSA level (r: 0.53–0.69, *p* < 0.05). The regression model significantly improves the explanatory value for PSA level (*p* < 0.05)
Song et al. [91]	Differentiate clinical significant PCa Gleason score ≥ 7 (3 + 4) from indolent disease	mpMRI	AUC on training, validation, and test dataset achieved results of 0.838, 0.814, and 0.824, respectively
Castillo et al. [92]	Differentiate high-grade versus low-grade lesions	mpMRI	The three single-center models obtained a mean AUC of 0.75, outperforming expert radiologist
Li et al. [107]	Prediction of clinically PCa Gleason score ≥ 7 (3 + 4)	Biparametric mpMRI	Both the radiomics model (AUC: 0.98) and the clinical–radiomics combined model (AUC: 0.98) achieved greater predictive efficacy than the clinical model (AUC: 0.79)
Li, Q et al.	Detection of clinically significant PCa Gleason score ≥ 7 (3 + 4)	MRI	Built a linear classifier model on these semantic traits and related to pathological outcome to identify clinically significant tumors. The discriminatory ability of the predictors was tested using cross-validation method randomly repeated and ensemble values were reported
Bonekamp et al. [108]	Compare radiomics and mean ADC for characterization of prostate lesions (Gleason grade group ≥ 2)	MRI	Comparison of the area under the AUC for the mean ADC (AUC_global_ = 0.84; AUC_zone-specific_ ≤ 0.87) vs. the RML (AUC_global_ = 0.88, *p* = 0.176; AUC_zone-specific_ ≤ 0.89, *p* ≥ 0.493)
Bleker et al. [109]	Identification of clinically significant peripheral zone PCa Gleason score ≥ 7 (3 + 4)	mpMRI	Combined model T2w and DWI images through an auto fixed VOI with AUC 0.870 (95% CI 0.980–0.754)
Radiomics and detection of ECE			
Losnegård et al. [110]	Prediction of extraprostatic extension in non-favorable intermediate- and high-risk prostate cancer patients	mpMRI	Best AUC for extraprostatic extension prediction models used in combination (MSKCC + radiology + radiomics) 0.80
Ma et al. [94]	Identification of PCa ECE	mpMRI	AUC of 0.902 and 0.883 in the training and validation cohort, respectively. Outperforming the radiologists results (AUC range 0.600–0.697), (75.00% vs. 46.88–50.00%, all *p* < 0.05), respectively
Ma et al. [93]	Identification of PCa ECE	mpMRI	AUC of 0.906 and 0.821 for the training and validation datasets, respectively
Cysouw et al. [111]	Prediction of lymphovascular invasion nodal or distant metastasis and Gleason score	(^18^F)DCFPyL PET	Lymphovascular invasion (AUC 0.86 ± 0.15, *p* < 0.01), nodal or distant metastasis (AUC 0.86 ± 0.14, *p* < 0.01), Gleason score (0.81 ± 0.16, *p* < 0.01), and ECE (0.76 ± 0.12, *p* < 0.01)

ADC = apparent diffusion coefficient; AUC = area under the curve; DNA = deoxyribonucleic acid; DRE = digital rectal examination; DWI = diffusion-weighted imaging; ECE = extracapsular extension; GLM = generalized linear model regression; LASSO = least absolute shrinkage and selection operator; mpMRI = multiparametric magnetic resonance imaging; PIRADS v2 = prostate imaging reporting and data system version 2; PSA = prostate-specific antigen; PZ = peripheral zone; SAVR = surface area-to-volume ratio; T2w = T2-weighted; TZ = transitional zone; VOI = volume of interest.

**Table 3 ijms-22-09971-t003:** Compilations of studies on the association of imaging and genomics.

Biomarker	Description	Test Source	Analysis	Study	Results
Prostate cancer antigen 3	Prostate-specific mRNA quantification	Prostate biopsy	Negative prior biopsy	De Luca et al. [138]	Significant association between PCA3 score and PI-RADS grade groups 3, 4, and 5 (*p* = 0.006)
			Two negative prostate biopsies	Alkasab et al. [139]	PCA3 not statistically correlated with PCa diagnosis (*p* = 0.128) and PCA3 associated with high-grade PCa at final pathology (*p* = 0.0435)
			No prior biopsy	Fernstermaker et al. [140]	PCA3 associated with MRI suspicion score of 2 and 3 (*p* = 0.004), not 4 and 5 (*p* = 0.340)
			Negative prior biopsy	Perlis et al. [141]	Normal PCA3 score gave a negative predictive value of 100% (*p* < 0.0001)
Decipher test^®^	22 RNA markers for prognosis and prediction of metastasis	RP or prostate biopsy	Low and intermediate PCa	Martin et al. [142]	Decipher^®^ biopsy genomic test was associated with Gleason grade group and it was independent of PIRADSv2 score
			Defining the favorable intermediate-risk prostate cancer	Falagario et al. [143]	Unfavorable intermediate-risk category (*p* < 0.001) and Decipher^®^ test (*p* = 0.012) were statistically significant predictors of adverse pathology; mpMRI did not maintain statistical significance (*p* = 0.059)
			Prediction of BCR	Jambor et al. [144]	Decipher^®^ genomic score and mpMRI could not improve predictive performance of biochemical recurrence compared with the individual use of these features
			mpMRI could predict aggressive prostate cancer features	Beksac et al. [145]	Association of Decipher^®^ score was significantly with lesion size (*p* = 0.03), PIRADS score (*p* = 0.02) and extraprostatic extension (*p* = 0.01)
			Correlation between MRI phenotypes of PCa as defined by PI-RADS v2 and Decipher	Purysko et al. [146]	MRI-visible lesions had higher Decipher^®^ scores than MRI-invisible lesions (*p* < 0.0001); some lesions classified as intermediate/high risk by Decipher^®^ are invisible on MRI
			BCR and adverse pathology prediction	Li et al. [45]	New imaging-based nomogram; AUC (0.71, 95% CI 0.62–0.81) better than Decipher^®^ AUC (0.66, 95% CI 0.56–0.77) and prostate cancer risk assessment (CAPRA) score AUC (0.69, 95% CI 0.59–0.79)
Oncotype Dx test^®^	5 reference genes and 12 cancer genes generating a genomic prostate score (GPS)	Prostate biopsy	Association between mpMRI and Oncotype Dx test^®^GPS	Leapman et al. [147]	GPS differences among MRI categories for patients with Gleason pattern 3 + 4 (*p* = 0.010), not in Gleason pattern 3 + 3
			GPS to predict adverse pathology	Salmasi et al. [148]	GPS is a significant predictor for adverse pathology (*p* < 0.001)
ConfirmMDx^®^	Alterations in DNA methylation	Prior negative biopsies	mpMRI PIRADS score lesions after ConfirmMDx^®^ sampling	Artenstein et al. [149]	Negative ConfirmMDx^®^ test is in accordance with negative MRI results (71.4%). ConfirmMDx^®^ sampling may be useful as a fusion-targeted biopsy rather than systematic biopsy
Prolaris test^®^	46-mRNA genomic test	Prostate biopsy	Associations between MRI and the expression levels of cell cycle genes	Wibmer et al. [150]	In the RP subgroup, ECE on MRI (*p* ≤ 0.001–0.001) and cycle genes risk scores (*p* = 0.049) were significantly associated with Gleason score 4 + 3 or higher, ECE and lymph node metastases

AUC = area under the curve; BCR = biochemical recurrence; DNA = deoxyribonucleic acid; ECE = extracapsular extension; GPS = genomic prostate score; mpMRI = multiparametric magnetic resonance imaging; mRNA = micro ribonucleic acid; PCA3 = prostate cancer antigen 3; PIRADS v2 = prostate imaging reporting and data system version 2; RP = radical prostatectomy.

**Table 4 ijms-22-09971-t004:** Overview of radiogenomic literature on prostate cancer.

Reference	Molecule Studied	Imaging Performed	Results	Approach	Method
McCann et al. [124]	PTEN	MRI	Perfusion imaging contrast uptake, T2-weighted signal-intensity skewness	Classical	Radiomic
Stoyanova et al. [158]	General gene expression	MRI	Radiomic signatures	Classical	Radiomic
Renard-Penna et al. [119]	RNA expression signature derived from cell cycle proliferation genes (Prolaris^®^)	mpMRI	Correlation with Gleason score (r = 0.199, *p* = 0.04) and PIRADS sum score (r = 0.26, *p* = 0.007)	Classical	Radiomic
Jamshidi et al. [125]	Whole-exosome DNA sequencing	mpMRI	No statistically significant linear correlation between individual mutations and mpMRI imaging parameters or PIRADS scores (*p* = 0.3)	Classical	Radiomic
Houlahan et al. [130]	Small nucleolar RNAs	mpMRI	Elevated snoRNA abundance may be a novel hallmark of nimbotic tumors (AUC: 0.87; 95%CI: 0.75–0.99)	Classical	Radiomic
Li P et al. [159]	Differentially expressed genes	MRI	MRI visibility (AUC: 0.86), progression-free survival HR = 2.53 (1.55–4.11), *p* < 0.001 BCR-free survival HR = 1.3 (1.04–1.63), *p* = 0.021	Classical	Radiomic
Eineluoto et al. [160]	PTEN and ERG	MRI	MRI-invisible lesions had less PTEN loss and ERG-positive expression compared with patients with MRI-visible lesions (17.2% vs. 43.3%, *p* = 0.006; 8.6% vs. 20.0%, *p* = 0.125)	Classical	Radiomic
Hectors et al. [161]	40 gene expression signatures plus Decipher^®^	MRI	Prediction of Gleason score of 8 or greater (AUC 0.72) and prediction of a Decipher^®^ score of 0.6 or greater (AUC 0.84).	Classical	Radiomic
Li L et al. [162]	Decipher^®^	MRI	Model outperformed the prediction using PIRADS v2 (AUC = 0.67), and comparable performance with Gleason grade group (AUC = 0.80)	Classical	Radiomic
Sun et al. [163]	Full transcriptome genetic profiles	mpMRI	Weak association of mpMRI features and hypoxia gene expression (*p* < 0.05).	Classical	Radiomic
Fischer et al. [27]	Gene and miRNA expression (Alanyl membrane aminopeptidase, microRNA-mir-217, mir-592, mir-6715b)	mpMRI	T2c and T3b prostate cancer stages being highly correlated with aggressiveness on related imaging features (average r = ± 0.75)	Classical	Radiomic
Wibmer et al. [150]	Prolaris^®^ test	MRI	ECE on MRI had significantly higher mean cell cycle risk score (reader 1: 3.9 vs. 3.2, *p* = 0.015; reader 2: 3.6 vs. 3.2, *p* = 0.045)	Classical	Radiomic
Vander-Weele et al. [164]	PTEN	mpMRI	Imaging uptake parameters showing mathematical correlation with PTEN expression (r = 0.25, *p* < 0.1 and r = 0.43, *p* < 0.01), and T2w unevenness also showed some correlation tendency (r = −0.25, *p* < 0.1)	Classical	Radiomic
Switlyk et al. [165]	PTEN	MRI	ADC was negatively correlated with Gleason score (*p* = 0.001) and tumor size (*p* = 0.023)	Classical	Radiomic

ADC = apparent diffusion coefficient; AUC = area under the curve; DNA = deoxyribonucleic acid; ECE = extracapsular extension; ERG = ETS-related gene; mpMRI = multiparametric magnetic resonance imaging; miRNA = micro ribonucleic acid; PIRADS = prostate imaging reporting and data system; PTEN = phosphatase and tensin homolog; T2w = T2-weighted.

**Table 5 ijms-22-09971-t005:** Advantages and limitations of radiogenomics compared to actual management of PCa risk stratification.

Radiogenomics	Advantages	Limitations
	Could provide accurate imaging biomarkers, substituting for genetic testing with lower cost [179]	Lack of prospective studies [6]
	AI and deep learning using big public databases with genomics and imaging features will be used to develop computer-aided tools for clinical practice translation [27]	Image acquisition for defining and contouring the regions of interests need expert radiologists [26]
	Automatic and semi-automatic computer designed software used to reduce drawbacks (lack of standardization, imaging and reporting protocols which differ significantly among institutions) [189]	Significant time used for proper manual delineation [179]
	Radiomics/radiogenomics biomarkers may predictrisk and outcomes and may be used to personalize treatment options [179]	High inter-observer variability in reading and segmenting regions of interest [180]
	Insights into the tumor genome requires biopsies, an invasive procedure that may increase patient morbidity. Radiogenomics can predict tumor genomic alterations [26]	Lack of repeatability and reproducibility—no standardization—different acquisition protocols, scanners and radiomic studies [194,195]
	Availability of whole-tumor information with aradiomics-based approach that can providepredictive and prognostic data [196]	Matching the data from whole-genome sequencing with imaging data is difficult due to different patient characteristics and imaging protocols [179]

## Data Availability

Not applicable.

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
