# Peer review of "Prostate Cancer Radiogenomics—From Imaging to Molecular Characterization"

_ijms, 2021, doi:10.3390/ijms22189971_

Round 1

Reviewer 1 Report

The manuscript is interesting but long, and in some parts redundant. Please shorten it.

I have the following comments:

-INTRODUCTION. it would be useful to briefly explain the concept of genomics and the state of the art regarding prostate cancer.

In my opinion, the sentence “Radiogenomics have been 56 studied in only a few cancers, like glioblastoma [5–8], breast cancer [9–12], renal cancer 57 [13–15] and other common neoplasms summarized in review by Shui et al. [16]” lines 57-58, would be best suited in the last part of the paragraph before the aim of the study.

The aim of the study should be rephrased to better underline the useful of this work.

- RESULTS. I would suggest using a PRISMA flow diagram (Moher D (2009) Preferred reporting items for systematic reviews and meta-analyses: the PRISMA statement. Annals of Internal Medicine 151:264).

How many articles did you find in Pubmed, Scopus and Web of Science?

2.2 “Radiomics in prostate cancer management”. The first sentence would be best suited in paragraph 2.1, to avoid redundance.

3.3.2. “Radiogenomics”. The sentence reported in lines 311-319 should be moved to the introduction.

“Radiogenomics in prostate cancer management”: Lines 321-331 are redundant.

- DISCUSSION. In this form is only a summary of the result. Please, rephrase it.

- MATERIAL AND METHODS What type of restrictions were used in the search? (e.g. articles only in English, exclusion of review or conference paper, etc).

How many people performed the search? Manually or with EndNote or other software?

Lines 464-467 The aim of the study must not be declared in this section.

- CURRENT CHALLENGES/LIMITATIONS AND FUTURE PERSPECTIVES In my opinion this section could be integrated in the discussion.

What are the limits of this review?

Author Response

We want to thank the reviewer for the suggestions made in order to give the paper a highly academic degree suitable for the readers.

Q: The manuscript is interesting but long, and in some parts redundant. Please shorten it.

A: We thank the reviewer for suggesting and had performed changes to all possible redundant parts and we had shortened it as possible.

I have the following comments:

Q: -INTRODUCTION. it would be useful to briefly explain the concept of genomics and the state of the art regarding prostate cancer.

A: We thank the reviewer for suggesting a more extensive explanation of current state of art of genomics in PCa. We have added a new paragraph at Line 84-88.

“Being the latest studied in PCa genomics, the concept of heterogeneity of PCa, the intratumoral modifications, clonal and subclonal alterations, microheterogeneity, macroheterogeneity, the multifocal nature of PCa and the inter-tumoral heterogeneity need to be matched between imaging and molecular pathology for establishing the clinical implications”

Q: In my opinion, the sentence “Radiogenomics have been 56 studied in only a few cancers, like glioblastoma [5–8], breast cancer [9–12], renal cancer 57 [13–15] and other common neoplasms summarized in review by Shui et al. [16]” lines 57-58, would be best suited in the last part of the paragraph before the aim of the study.

A: We thank the reviewer for pointing out the better positioning of this paragraph. We made the suggested modification. Line 95-97

Q: The aim of the study should be rephrased to better underline the useful of this work.

A: We thank the reviewer for the suggestion. We added more explanations for the value of the article. Line 107-113

Q: - RESULTS. I would suggest using a PRISMA flow diagram (Moher D (2009) Preferred reporting items for systematic reviews and meta-analyses: the PRISMA statement. Annals of Internal Medicine 151:264).

A: We thank the reviewer for the suggestion. We changed the flow diagram accordingly.

Q: How many articles did you find in Pubmed, Scopus and Web of Science?

A: 1066 in Pubmed, Google Scholar and Web of Science, last search in June 2021, using radiogenomics, radiomics, prostate cancer and MRI words search criteria. We have focused on papers from the last five years.

Q: 2.2 “Radiomics in prostate cancer management”. The first sentence would be best suited in paragraph 2.1, to avoid redundance.

A: We thank the reviewer for the suggestion and we adapted the sentence in the suggested paragraph. Line 123-125

Q: 3.3.2. “Radiogenomics”. The sentence reported in lines 311-319 should be moved to the introduction.

A: We thank the reviewer for the suggestion and we moved the paragraph into the introduction.

Line 98-106

Q: “Radiogenomics in prostate cancer management”: Lines 321-331 are redundant.

A: We thank the reviewer for noticing the redundancy and we have erased the paragraph.

Q: - DISCUSSION. In this form is only a summary of the result. Please, rephrase it.

A: We thank the reviewer for the suggestion and we had made changes accordingly. Line 487-493

- MATERIAL AND METHODS What type of restrictions were used in the search? (e.g. articles only in English, exclusion of review or conference paper, etc).

Q: How many people performed the search? Manually or with EndNote or other software?

A: Two people performed the search and it has been done manually.

Q: Lines 464-467 The aim of the study must not be declared in this section.

A: We thank the reviewer for the suggestion and we had made changes accordingly.

Line 107-113 rephrased.

Q: - CURRENT CHALLENGES/LIMITATIONS AND FUTURE PERSPECTIVES In my opinion this section could be integrated in the discussion.

A: We thank the reviewer for the suggestion and have integrated the section into the discussion section.

Q: What are the limits of this review?

Line 493-497 Some limitations of this review have to be addressed. Only a few databases and online search engines were used to retrieve relevant data. The search for the above topic was performed manually by two independent researchers. Probably scrutinizing more databases and registries using a software search may have provided more relevant data for the subject.

We have included this paragraph into the main text of the review.

Reviewer 2 Report

The authors should include a citation analysis in their survey.

The most influential articles should be further analyzed 

Author Response

We want to thank the reviewer for the suggestions made in order to give the paper a highly academic degree suitable for the readers.

Q: The authors should include a citation analysis in their survey.

A: We thank the reviewer for the suggestion and have integrated an analysis of citations used for radiogenomics section after each presented study.

Q: The most influential articles should be further analyzed.

A: We thank the reviewer for the suggestion. For the studies of radiomics in prostate cancer we had performed a brief reminder, and incorporated them in Table 2. The most influential radiogenomic articles were analyzed in that section. We adapted the text and stated:

Line 157-161  “Although it is beyond the purpose of this review to focus on the radiomics technical terms, we have had briefly reminded them because they represents the starting point in this research field. Articles that studied radiomics in PCa are just briefly reminded and not analyzed in detail and we had them incorporated in a table, along with their clinical outcomes and results.”

Reviewer 3 Report

In this review, Tataru et al. present a comprehensive summary of the current literature on quantitative and qualitative results from well-designed studies for the diagnosis, treatment, and follow-up of prostate cancer based on radiomics, genomics, and radiogenomics. In fact, radiomics/radiogenomics is an important area, and I judge this review to be of significance to researchers and medical practitioners, especially those specializing in prostate cancer. I think it is well organized, but I recommend that the authors pay attention to the following points and revise it before publishing the manuscript in the journal.

[Major]

I believe that there are two particular problems in the clinical application of radiomics and radiogenomics. The first is that the process of analysis is complex, which makes it a black box and difficult to interpret by humans. This issue is discussed in the limitations section of this review, so I think it is fine to leave it as it is now. However, since the field of explainable AI (XAI) is currently developing, I think it is fine to include the following references and other explanations for the benefit of readers.

*Papadimitroulas et al., “Artificial intelligence: Deep learning in oncological radiomics and challenges of interpretability and data harmonization “, Physica Medica, 83, 108-121, 2021. DOI: 10.1016/j.ejmp.2021.03.009

*Yang et al., "Probabilistic radiomics: Ambiguous diagnosis with controllable shape analysis", MICCAI, 658-666, 2019. DOI: 10.1007/978-3-030-32226-7_73

Another important issue is the problem of domain shift, which is often observed in multicenter studies. In particular, I consider the following to be one of the most important papers.

Pooch et al., “Can we trust deep learning models diagnosis? The impact of domain shift in chest radiograph classification”, arXiv:1909.01940 (2019)

Currently, there are attempts to solve the problem of domain shift by using domain adaptation and fine-tuning methods, so I believe that the authors may refer to the following papers if necessary.

*Ackaouy et al., “Unsupervised Domain Adaptation With Optimal Transport in Multi-Site Segmentation of Multiple Sclerosis Lesions From MRI Data“, Front. Comput. Neurosci, 14:19, 2020. DOI: 10.3389/fncom.2020.00019

*Takahashi et al., " Fine-Tuning Approach for Segmentation of Gliomas in Brain Magnetic Resonance Images with a Machine Learning Method to Normalize Image Differences among Facilities", Cancers (Basel), 13(6):1415, 2021. DOI: 10.3390/cancers13061415

[Minor]

  1. As for Line 320, I think it should be numbered as "3.3.2.1. Radiogenomics in prostate cancer management" as it gives a sense of discomfort.

  1. There should be a space between Line 524 and 525.

Author Response

We want to thank the reviewer for the suggestions made in order to give the paper a highly academic degree suitable for the readers.

In this review, Tataru et al. present a comprehensive summary of the current literature on quantitative and qualitative results from well-designed studies for the diagnosis, treatment, and follow-up of prostate cancer based on radiomics, genomics, and radiogenomics. In fact, radiomics/radiogenomics is an important area, and I judge this review to be of significance to researchers and medical practitioners, especially those specializing in prostate cancer. I think it is well organized, but I recommend that the authors pay attention to the following points and revise it before publishing the manuscript in the journal.

[Major]

Q: I believe that there are two particular problems in the clinical application of radiomics and radiogenomics. The first is that the process of analysis is complex, which makes it a black box and difficult to interpret by humans. This issue is discussed in the limitations section of this review, so I think it is fine to leave it as it is now. However, since the field of explainable AI (XAI) is currently developing, I think it is fine to include the following references and other explanations for the benefit of readers.

*Papadimitroulas et al., “Artificial intelligence: Deep learning in oncological radiomics and challenges of interpretability and data harmonization “, Physica Medica, 83, 108-121, 2021. DOI: 10.1016/j.ejmp.2021.03.009

*Yang et al., "Probabilistic radiomics: Ambiguous diagnosis with controllable shape analysis", MICCAI, 658-666, 2019. DOI: 10.1007/978-3-030-32226-7_73

 A: We thank the reviewer for the suggestion and we have added text to explain the new developments in AI and radiomics that better explain the potential future of DNNs applications.

Q: Another important issue is the problem of domain shift, which is often observed in multicenter studies. In particular, I consider the following to be one of the most important papers.

Pooch et al., “Can we trust deep learning models diagnosis? The impact of domain shift in chest radiograph classification”, arXiv:1909.01940 (2019)

 A: We thank the reviewer for the suggestion and we have added the article in the reference list.

Q: Currently, there are attempts to solve the problem of domain shift by using domain adaptation and fine-tuning methods, so I believe that the authors may refer to the following papers if necessary.

*Ackaouy et al., “Unsupervised Domain Adaptation With Optimal Transport in Multi-Site Segmentation of Multiple Sclerosis Lesions From MRI Data“, Front. Comput. Neurosci, 14:19, 2020. DOI: 10.3389/fncom.2020.00019

*Takahashi et al., " Fine-Tuning Approach for Segmentation of Gliomas in Brain Magnetic Resonance Images with a Machine Learning Method to Normalize Image Differences among Facilities", Cancers (Basel), 13(6):1415, 2021. DOI: 10.3390/cancers13061415

  A: We thank the reviewer for the suggestion and we have added these articles in the reference list.

[Minor]

  1. As for Line 320, I think it should be numbered as "3.3.2.1. Radiogenomics in prostate cancer management" as it gives a sense of discomfort.

We thank the reviewer for the suggestion and we had modified the number of the paragraph.

  1. There should be a space between Line 524 and 525.

We thank the reviewer for the suggestion and we had added a space between paragraphs.